# Kiukainen Culture Site Locations—Reflections from the Coastal Lifestyle at the End of the Stone Age

Janne Soisalo [1,*] and Johanna Roiha [2]

1 Department of Cultures, Faculty of Arts, University of Helsinki, 00014 Helsinki, Finland
2 Faculty of Agriculture and Forestry, University of Helsinki, 00014 Helsinki, Finland
* Correspondence: janne.soisalo@helsinki.fi

**Abstract:** The Kiukainen culture constitutes a poorly known phase at the end of the Stone Age in Finland, approximately 2500–1800 cal. BC. It is best known for its pottery, and most of the finds are from the coastal area of the Baltic Sea between Helsinki and Ostrobothnia. Previous research on the culture was done several decades ago, so this study aims to define the geographical distribution of the sites known thus far and discuss the landscape around the settlement sites. Creating an overall view of the culture and lifestyle of the people is also an important part of the study. First, it focuses on different collections of Kiukainen pottery and then maps the location of all the sites where pottery has been found. For the landscape visualizations, three different areas were chosen for closer evaluation. Elevation models were, then, used to visualize the Stone Age coastal landscape. Altogether, we identified 99 settlement sites with a confirmed connection to Kiukainen culture. One common feature of the locations is a connection to the sea. The sites are located in various types of environments, but they all have easy access to seafaring and good landing possibilities from the sea.

**Keywords:** Stone Age; Kiukainen culture; pottery; settlement site; Baltic Sea; shoreline modelling; landscape archaeology; coastal changes

## 1. Introduction

The Kiukainen culture was a coastal Neolithic culture that existed on the southern and western coasts of Finland during approximately 2500–1800 cal. BC, starting at the beginning of the Final Neolithic and continuing until the Bronze Age. Its central distribution area extends from southern Ostrobothnia to the Gulf of Finland near the Helsinki region (Figure 1), but only a few inland settlements have been discovered to date. Outside the actual core area, Kiukainen ceramics have been found in some known Neolithic settlement sites. The Kiukainen culture was a uniform cultural group in terms of pottery and stone artefacts, as well as in terms of living in a maritime environment, along the coastline of the Baltic Sea. It was first identified as a unique cultural form by the Finnish archaeologist Julius Ailio in 1909 [1], and since then, the Kiukainen culture has, periodically, been the subject of more focused research. The last in-depth study dealing with the culture is Carl Fredrik Meinander's 1954 book *Die Kiukaiskultur* [2].

The Kiukainen culture was preceded by the pan-European Corded Ware Culture, 2900–2200 cal. BC, which spread to Finland from the southern Baltic region by two local cultural groups. They included the Pyheensilta group [3], which lived on the southern and western coasts of Finland during approximately 3200–2400 cal. BC, and the Pöljä group, which mainly inhabited the inland and the coast of Ostrobothnia during 3200–2500 cal. BC [4]. Scholars believe that the Kiukainen culture arose as a result of the diffusion of these different populations and cultural forms, but the populations also had significant connections with contemporary Scandinavia at the time [2,5]. The diffusion can be seen in material culture. It has recently been suggested that the Kiukainen culture ended with a period of desolation and cultural interruption before the Bronze Age began on the coasts

of Finland [6]. However, the continued presence of people in the Bronze Age in many settlement sites and distribution areas used by the Kiukainen culture speaks against this theory. The beginning of the Bronze Age eventually occurred gradually due to strong Scandinavian connections and cultural influence, but from an archaeological standpoint, the change of eras is a time of few discoveries.

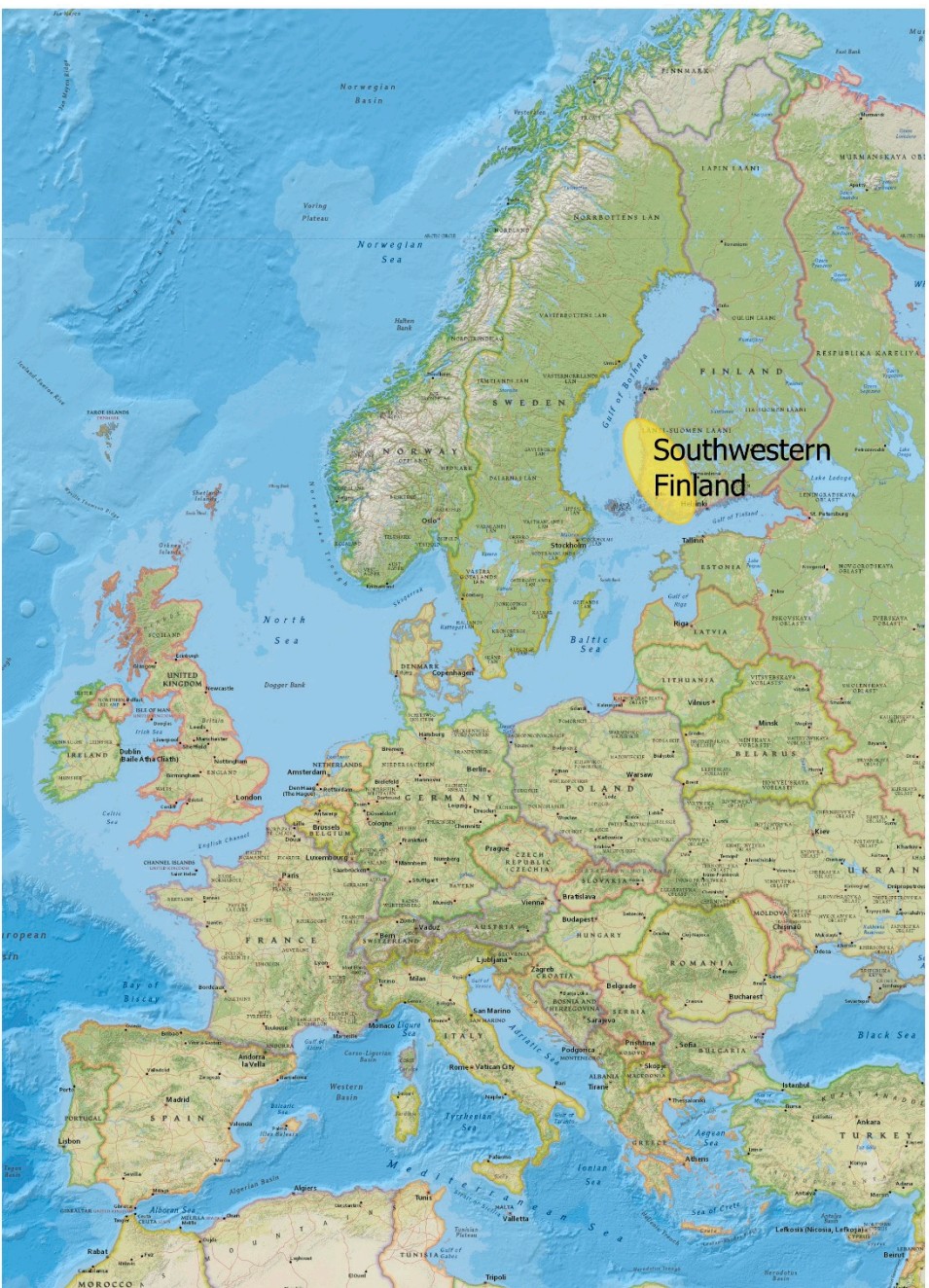

**Figure 1.** Southwestern Finland, highlighted in yellow. The central distribution area of Kiukainen culture extends from southern Ostrobothnia to the Gulf of Finland near the Helsinki region.

Kiukainen culture settlements have typically been discovered on sandy beaches by the sea, with the largest concentrations being in the inner archipelago and in the estuaries of rivers that flow into the sea, such as in today's Turku city region, at the mouth of the Kokemäenjoki River in Harjavalta and Lappfjärd in Kristiinankaupunki. In many places, the settlements have been several hectares in size, and inhabitation continued in such places for hundreds of years. However, it is difficult to determine, without available datings,

whether the use of the site was continuous or whether nearby settlement sites were used at different times. In addition, the use of the sites continued in many places into the Bronze Age, which has often made it difficult to date discoveries. Inhabitation along the coasts only became permanent, at latest, during the time of the Kiukainen culture, and scholars have suggested that the late settlements were no longer located on the sandy beaches next to the sea but on the other side of meadows, suitable, perhaps, for grazing, located between such former sites and the sea [7,8]. Settlement sites have also been discovered far out in the archipelago. These were possibly seasonal sites focused on marine fishing and hunting [9].

The connection between archaeological site locations and shoreline displacement has been a point of research interest in many studies in Finnish archaeology, e.g., [10–13]. Some more recent studies with more precise GIS materials and methods have been done in the last 15 years, as more open-access materials have become available [14]. Recent studies in geology have also begun to focus on land uplift and shoreline displacement since more precise GIS data is now available [15,16]. In 2001, researchers conducted an interesting study that modelled dwelling sites and sources of livelihood in the Espoo area near Helsinki [17]. The study concluded that Kiukainen sites in the Espoo area had a strong connection to marine resources, with sites being located on the seashore and islands. According to the study findings, change in the location of the settlements reflected major cultural changes during the formation of Kiukainen culture. However, it only included five Kiukainen sites, and the area of the study was quite limited, so statistical methods could not be used in the study. In Finland, more modern GIS analyses or methods have not yet been fully adapted to archaeology, but some basic studies have been done using, for example, interpolation methods [18] and cost surface analysis [19]. In Norway, a very interesting and relevant study was conducted in 2021 [20]. The research established that the peopling of the coastline in prehistory involved a series of active choices, and the main factors informing these decisions were good landing conditions and monitoring locations, followed by sufficient shelter from prevailing winds.

The first aim of this study is to define the geographical distribution of Kiukainen pottery and Kiukainen culture. Though some basic studies on Kiukainen culture were done decades ago, the results of those studies are outdated. The overall view of Kiukainen culture is indistinct, and more information is needed about the essence of Kiukainen culture. The geographical distribution of the culture has not been fully studied before, since the research focus, to date, has been more on individual sites or certain areas, as one study from the year 2001 points out [17]. The second aim of the study is to determine the types of landscapes or environments in which the settlement sites of Kiukainen sites were located. The choice of residence and local environment around the sites can give hints about the subsistence strategy of the culture. The third aim is to discuss the lifestyle of Kiukainen culture settlements based on research knowledge collected thus far. By lifestyle, we mean more than just subsistence strategy or nutrition. The term also includes, for instance, cultural contacts, traveling, social networks, and artefacts, which together constitute the mode of living of an individual or group. A broad perspective is important, and thus, this article highlights and especially discusses such an aspect.

The existing studies of individual archaeological sites and interpretations, based on only one site, are comparable to a study of finds without any context. Without a broader overall cultural picture in the background, the interpretations of individual sites remain weak and thin. From the fieldwork perspective, an overall picture is needed to provide more of a specific research focus during field studies. Knowledge about the geographical distribution of Kiukainen culture can also support future field studies because, most likely, many sites are still unidentified or undiscovered. Similar to artefacts, archaeological sites or monuments also have their own context, which is an idea that has inspired us in this research project.

## 2. Materials and Methods

Kiukainen ware forms its own uniform group that differs from preceding or contemporary pottery styles in the Northern Baltic Sea region. The vessels are thick-edged and rough-made, always flat-bottomed, and are usually straight-edged designs. Mild profiling also occurs sometimes. The vessels vary in size from small beakers to large storage vessels, but most are a few litres in size. Clay material was often mixed with crushed stone or sand, but organic temper or limestone was often used as well, causing the ceramics to be porous. Only the upper part of the surviving vessels is decorated. However, the lower undecorated parts often contain a textile imprint. The decoration consists of horizontal rows of pits, dots, lines, comb or ring stamps, and sometimes, spiral cord prints (Figure 2). Horizontal or vertical zig-zag lines are also typical. Though other decorations have also been found on vessels, the vessel is usually decorated only with pits and one other decorative element; for more, see [1,2,5].

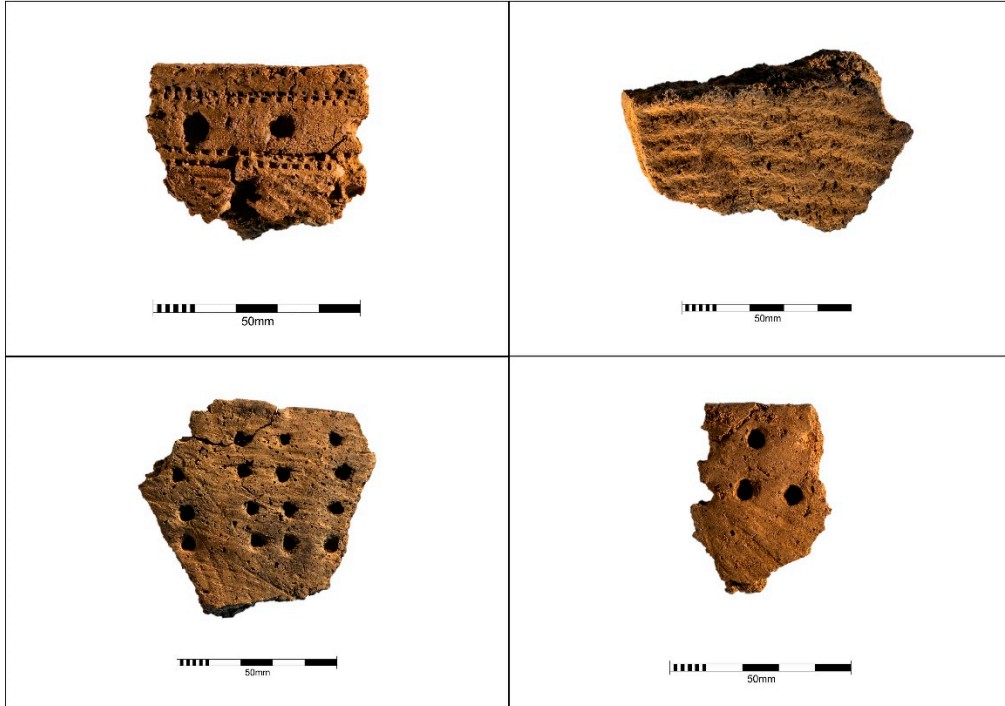

**Figure 2.** Kiukainen pottery, photo by Marjo Karppanen. Upper right bottom of a vessel with textile imprints; the rest are rim sherds.

The preliminary search for settlement sites representative of Kiukainen culture has been based on information provided in the Register of Ancient Monuments, previously published studies, and excavation reports. Based on this information, we have viewed all the finds from settlement sites that seem promising and have confirmed that they belong to the Kiukainen culture as a result of Kiukainen-type ware found at the sites. Most of the work has been done in the collections of the Finnish Heritage Agency, the Museum of Åland, and the Museum of Satakunta.

One challenge in defining the settlement sites is that little research has been done on them, and none of them have been fully excavated. Many of them have been found as part of archaeological surveys, meaning that, typically, only a small amount of material has been found, and it can be difficult to identify ceramics with certainty. The easily corroding and largely undecorated ceramics have also presented difficulties of their own because only decorated or otherwise clearly diagnostic pieces of pottery can be identified as Kiukainen with any degree of certainty.

After analysing and recognizing Kiukainen pottery, the coordinates of site locations confirming a connection to the culture were collected from the Register of Ancient Mon-

uments. The Finnish Heritage Agency maintains and updates the register. The register includes information about site type, location, possible dating, descriptions, and possible links to research reports. Sites also have a name and individual number code, which are used to list the sites. Every archaeological site in the register has coordinates in point format, and most of the sites have protected area definitions in polygon format. All information about archaeological sites in Finland is open-access form. The register can be found at the Finnish Heritage Agency's website, at its Cultural Environment service window (Kulttuuriympäristön palveluikkuna), but only in Finnish [21]. It is also possible to download the register for GIS use or else use it in GIS programs via open geographic information interfaces (VMS and VFS forms). After downloading the site register, it was possible to identify and list all sites where Kiukainen pottery has been found with the QGIS program. However, some sites excavated decades ago are not in the register, so some of those site locations are uncertain. Additionally, a few sites located outside the present borders of Finland (the Karelia area of Russia) were left out because the locations of those sites are uncertain. Sites located in the Åland Islands were identified using the Kulturarv website [22], updated by the Åland provincial government.

The mapping of the sites where Kiukainen ware has been found revealed some interesting site clusters. After examining the distribution results, three areas were chosen for closer evaluation and comparison. One factor in the choice was previous research history and knowledge of the sites in the area. For instance, some excavations of possible importance were done decades ago with poor documentation levels, while at other sites, the available research is quite limited and was only done at a small scale. Not all sites have confirmed dating since radiocarbon dating was never done on the finds. Those sites where the amount of Kiukainen pottery that was found was very small and other pottery types were dominant were considered too uncertain to compare. Comparison areas were chosen far from each other, where the landscape and topography are different. Many interesting sites are located near the city of Turku. The Turku city area was not studied as part of this research project, though, due to heavy land use and buildings.

Three areas that were chosen are the municipalities of Kemiönsaari, Harjavalta, and Kristiinankaupunki. Two nearby sites from the municipality of Nakkila were included in the Harjavalta study area because of the close geographical connection between them. To visualize the Stone Age shoreline, digital elevation models (elevation model 2 m) were downloaded from the open-data file service of the National Land Survey of Finland [23]. The elevation models are raster datasets that are based on laser scanning data, the point density of which is at least 0.5 points per square meter [24]. With the QGIS program, basic data visualization tools (unique values) were used to colourize the water blue to illustrate the shoreline. Information about changing sea levels during the Stone Age was collected from many different available sources, such as excavation reports and shoreline displacement chronologies. The Geological Survey of Finland provides open access to GIS data about the different soil types [25] in Finland. Unfortunately, the Geological Survey of Finland's most accurate soil type datasets do not cover the full Kemiönsaari area or Kristiinankaupunki area. As a replacement, the datasets from the Finnish Forest Centre were used to identify rocky areas. The Finnish Forest Centre datasets can be downloaded from its website [26] or used via open geographic information interfaces (VMS and VFS forms). The datasets include information about soil type in the forestry areas of Finland and the datatype area polygons. For background information and knowledge about the site (found on the Finnish Heritage Agency's webpages, specifically its cultural environment research reports), previous fieldwork history, such as excavation reports or survey reports, were also used.

## 3. Results

Altogether, we identified 99 sites with confirmed connections to the Kiukainen culture (Figure 3). The list of the sites can be found in Appendix A (Table A1). Uncertain cases, where the pottery could not be clearly identified, and those sites that have an inaccurate

location were left out of the results. The distribution of Kiukainen culture sites is strongly connected to the Stone Age shoreline of the Baltic Sea. The core area, where the number of sites and pottery finds is highest, is the shoreline between Espoo and Kristiinankaupunki. The results include only six inland sites where Kiukainen pottery could be identified. Three of those sites could be reached from the sea via the Kokemäenjoki River. The distribution map also revealed an approximately 80 km gap in shoreline colonialization between Pori and Kristiinankaupunki. The reason for the gap remains unclear, but it can also reflect a gap in field research history. The distribution map of Kiukainen culture can be considered, to some extent at least, to also reflect the general state of research on the Stone Age in Finland, as archaeological surveys have primarily focused on areas of changing land use around modern growth centres. Almost all archaeological surveys in Finland are done by commercial archaeology companies for different types of zoning and construction projects. Surveys are rarely done in areas that do not have active land use. It should also be strongly highlighted that the Kiukainen pottery findings are from sites that have been excavated. Those sites that have not been excavated but that are listed in the register after an archaeological survey are difficult to identify because only a very limited number of finds are collected during the survey. The total number of Kiukainen sites is most likely much higher, and the distribution map only reflects the current research situation.

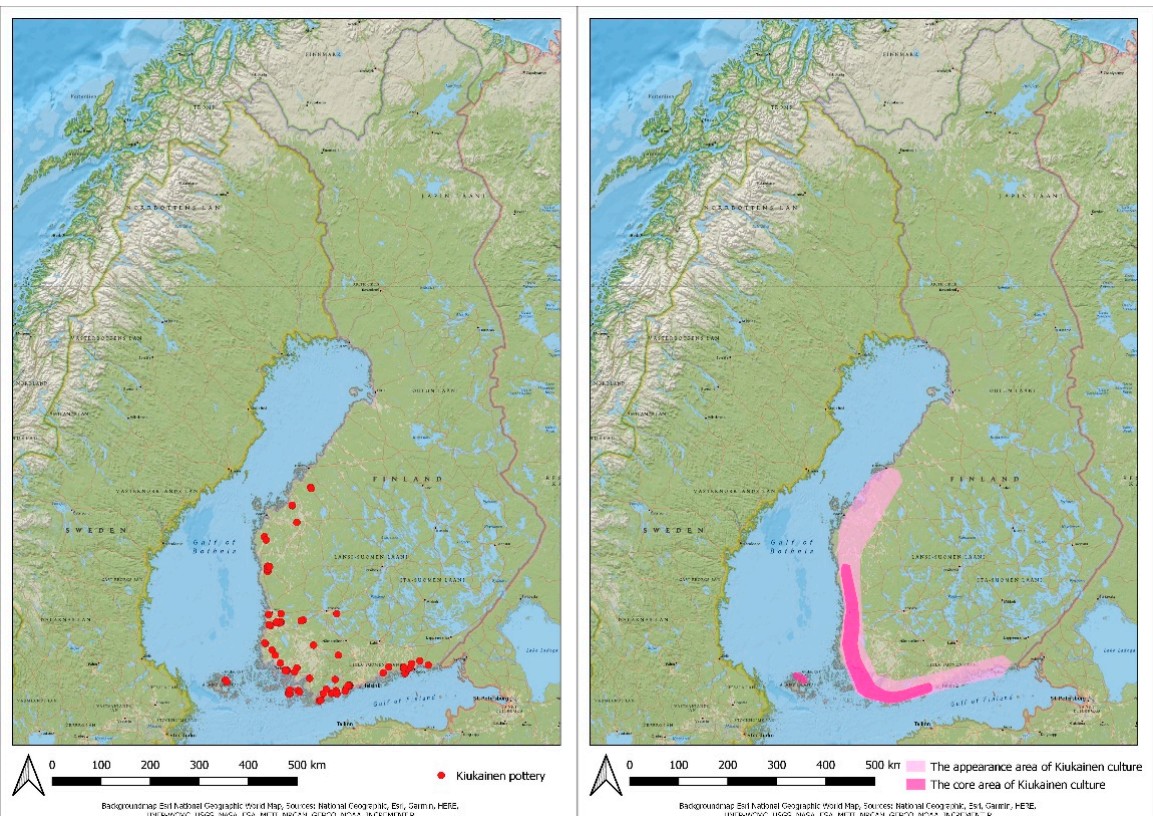

**Figure 3.** The distribution map of Kiukainen culture (**right**) where individual sites are marked with red dots. The heatmap of the Kiukainen culture (**left**). The heatmap was constructed by evaluating site density and also the number of pottery finds.

The three areas chosen for closer review, Kemiönsaari, Harjavalta, and Kristiinankaupunki, are located about 100 km apart from each other (Figure 4). Kemiönsaari, in the Southwest Finland region, is the southernmost of the sites, and it is also currently part of the archipelago. The Harjavalta area is in the middle of the Satakunta region, formerly part of the Western Finland Province. The northernmost review area is Kristiinankaupunki, in the Ostrobothnia region.

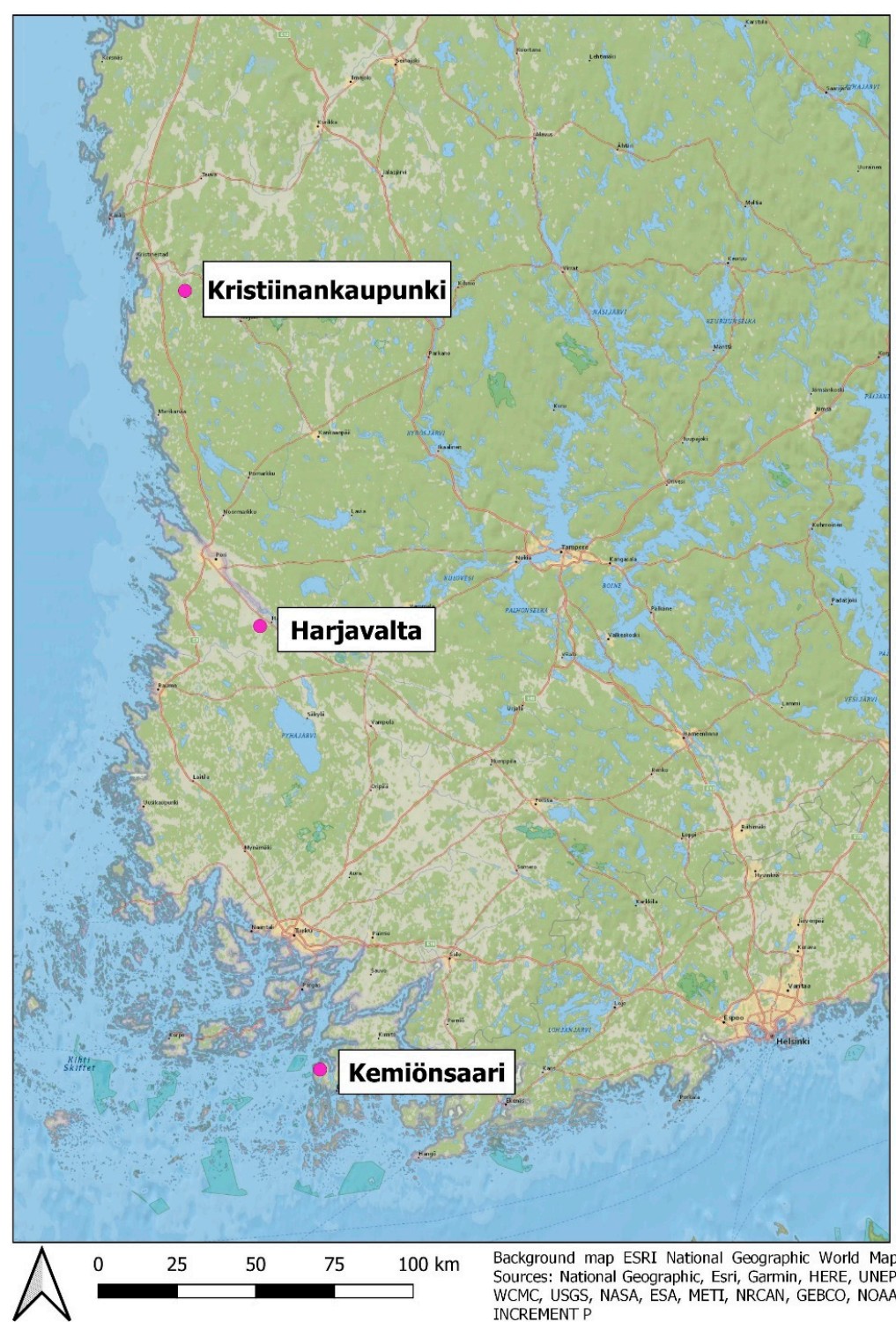

**Figure 4.** The Kiukainen culture areas that were selected for closer review.

　　　The Kiukainen sites in Kemiönsaari are located on large rocky islands in the archipelago area (Figure 5). Only the northernmost site of Näset was located on the shore of a smaller island. The sites are oriented towards the east because it afforded the best shelter from the western winds and better landing possibilities while navigating at sea (Figure 6). Since the sites are on islands, it is obvious that seafaring was quite familiar to the people of the Kiukainen culture. Landing on sandy beaches must have been easier than landing on a rocky shoreline, which could be one explanatory factor for site locations in the area.

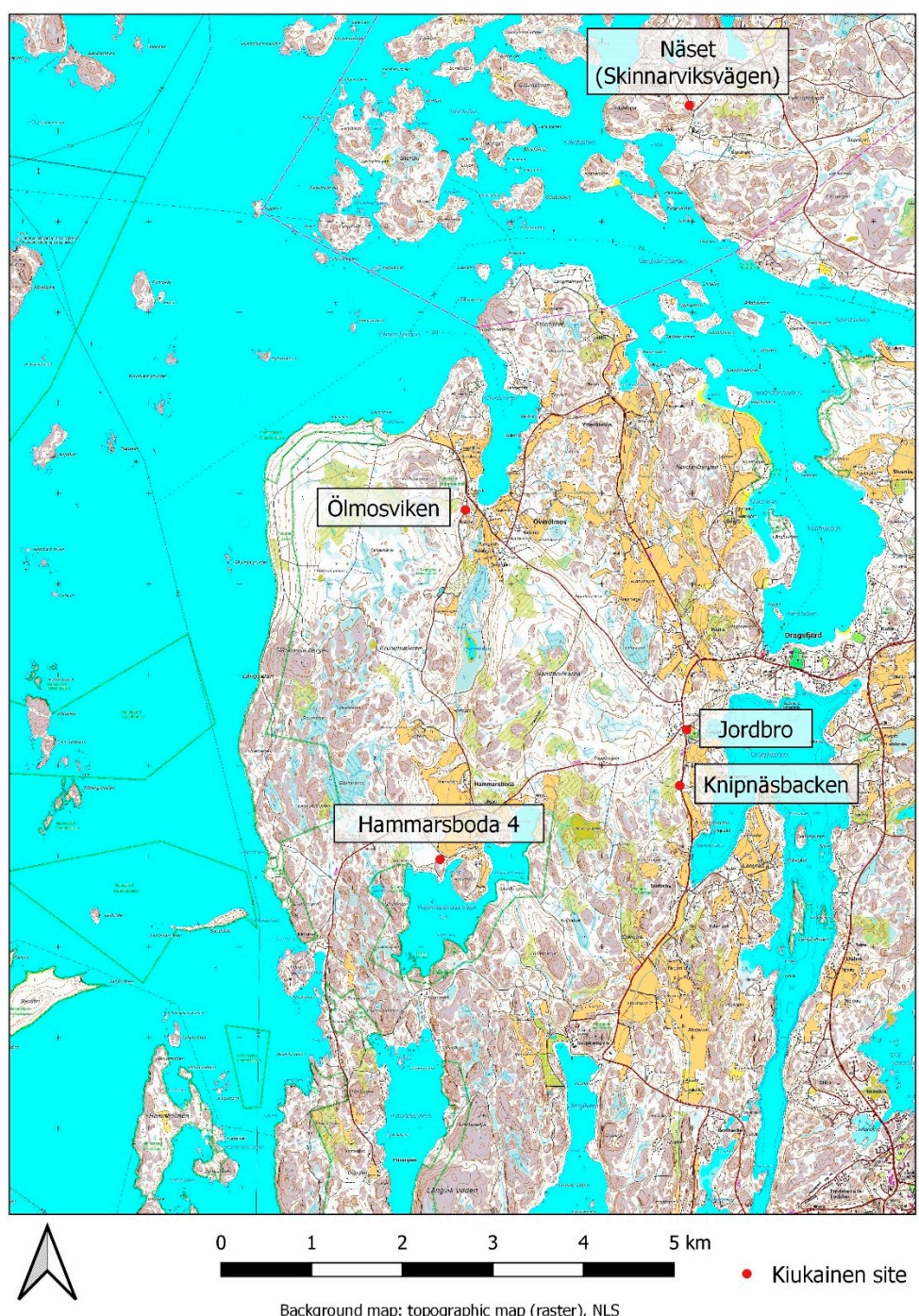

**Figure 5.** The locations of Kiukainen sites in the Kemiönsaari area. Grey areas in the background of the map are rocky areas, while the yellow areas are cultivated fields, and the light green or empty white areas in between are forests.

Archaeological excavations have been carried out at four of the settlement sites in the area. Jordbro and Knipängsbacken were partially excavated by C. F. Meinander in 1947, while small excavation was done in Hammarsboda by the University of Turku in 1991, and excavations were done at the Ölmosviken site in 2017–2021 [2,27,28]. Based on the research and C14 dating, the settlement sites may have been used at different points in time, but Ölmosviken shows signs of habitation for hundreds of years between 2300 and 1800 cal. BC. Jordbro is possibly younger than the other settlements, as five Bronze Age burial cairns have been discovered there, demonstrating a continuity of settlement to the Bronze Age.

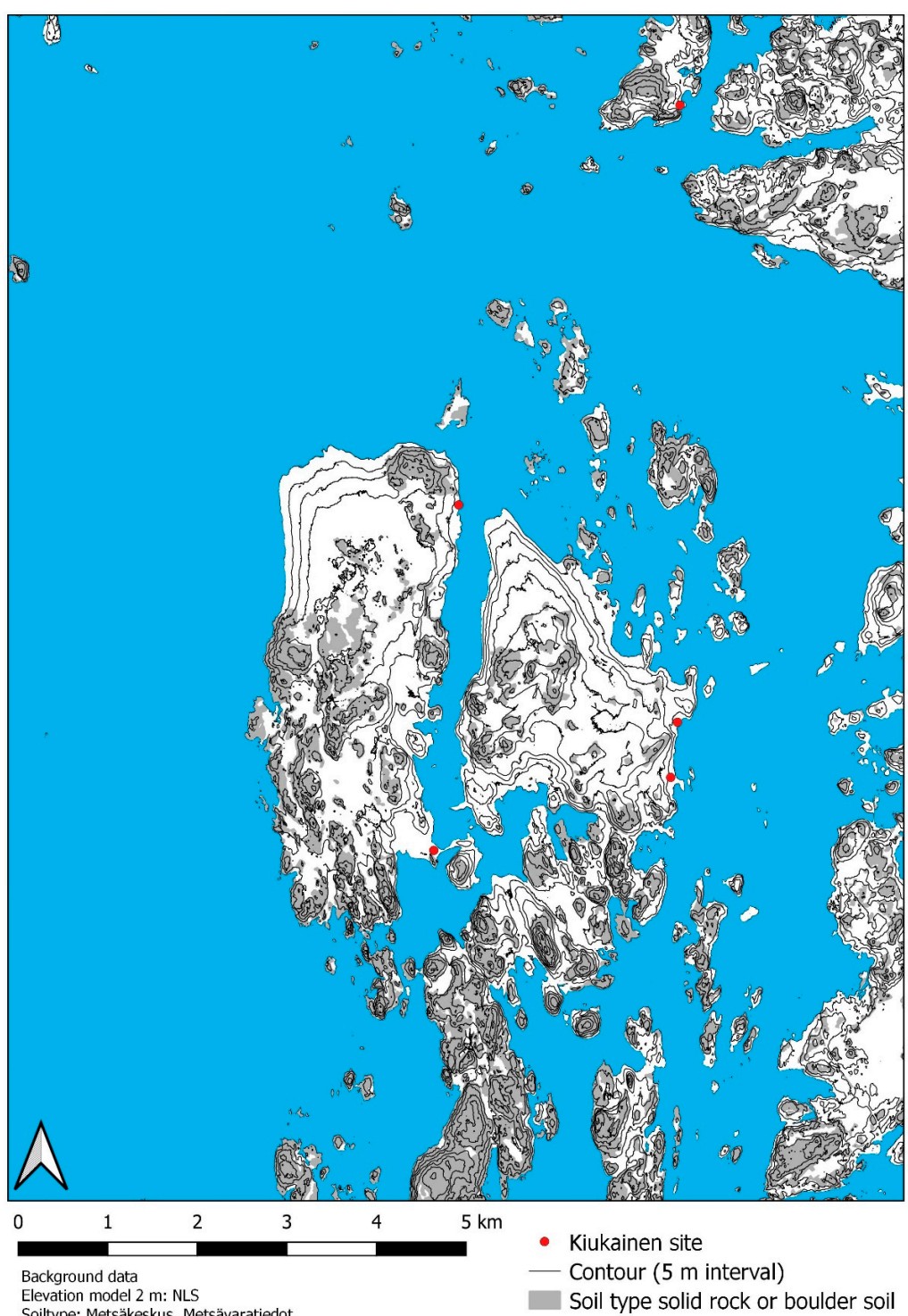

**Figure 6.** The locations of Kiukainen sites in the Kemiönsaari area and sea are visualized at 19 MASL. Areas with the soil type solid rock or boulder soil are visualized as grey.

The soil near the sites is mainly sand and gravel and, therefore, not particularly fertile, but on the other hand, larger barren-rock areas are located a little further away. Pollen analysis has been done at the nearby Söderbyträsket Lake, revealing that Pinus, Betula, and Alnus were the dominant types of trees during the period of Kiukainen culture. Additionally, Quercus, Populus, Tilia, Fraxinus, Ulmus, and Corylus grew nearby, which, together, accounted for about 20% of the vegetation [29]. Thus, except for the most barren areas, the area consisted mainly of deciduous forest, and the vegetation was lusher than

today. The first signs of cultivation are from the Bronze Age, 1210–1010 cal BC. [29], but the surroundings close to the residences would have been suitable, at least, for keeping goats and sheep already at the end of the Stone Age. So far, however, research has revealed no signs of such livestock practices, but the burned bone material is dominated by seal bones, at least at Ölmosviken [30]. A considerable number of the bones come from young individuals, which suggests that the catch took place in the spring and early summer.

The archipelago area was particularly favourable for seal hunting and fishing in the Stone Age, which, together with seabird hunting, were the most likely reasons for people moving to the area and for the establishment of settlements. The west and south sides of the island group would have given way to a wide and open sea, but the surroundings of the settlements consisted of sheltered archipelagos. This type of environment provided an abundance of fish, birds, and seals and, thus, plenty of food for people throughout the year. While information is unclear as to whether the sites were inhabited year-round, Ölmosviken contains traces of the dwelling pits. The pits probably originated from buildings partially dug into the ground, which would have been warm enough for people to live in during the cold seasons. On the mainland, the nearest large settlements would have been in the Turku region, about 45 km away and close to the sea, so the settlement of the area can also be connected to the marine fishing practiced by the communities that lived there.

The landscape in Harjavalta is quite different than in the Kemiönsaari area. In Harjavalta, the topography is a plane, and sites are located in small, forested areas near cultivated fields (Figure 7). The Kiukainen culture was discovered and named after the settlement site of Uotinmäki in the area at the beginning of the 20th century. In addition to Uotinmäki, archaeological excavations were carried out at Kaunismäki and Saamanmäki, the results of which have been presented in the book *Die Kiukaiskultur* [2]. Later, excavations were also done at the Lyytikänharju site, which nonetheless dates mainly to the time of the Pyheensilta group [31,32]. In recent years, residences named Kraakanmäki 1–3 have been investigated, with newer research results and C14 dates available from two of the excavations [33,34].

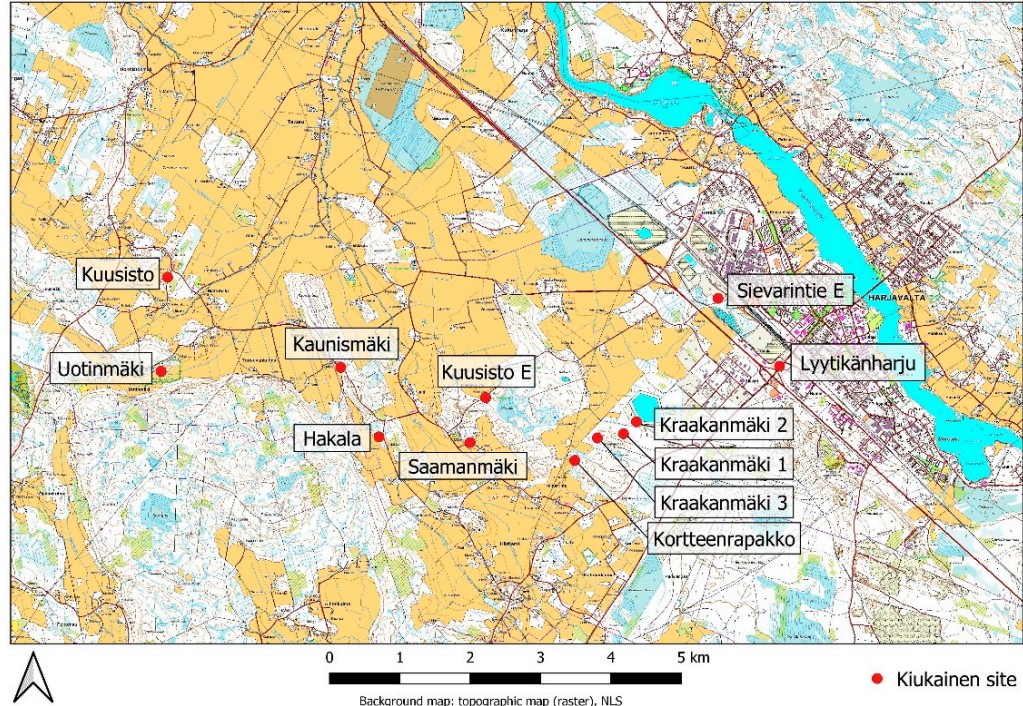

**Figure 7.** The locations of Kiukainen sites in the Harjavalta area. Grey areas in the background of the map are rocky areas, while the yellow areas are cultivated fields, and the light green or empty white areas in between are forests.

The settlements are located on the shores of a large and sheltered sea bay. The Kokemäenjoki River flowed into the bay, forming an estuary there (Figure 8). Due to the large flow of the river and the shallowness of the bay, the water in the bay has been brackish with very little salt. Many of the shorelines were probably lined with thick reeds. The area has been attractive, especially, in terms of fishing, as Kokemäenjoki River was well known for its salmon during historical times. The shallow reed banks have also attracted other fish and waterfowl to the area.

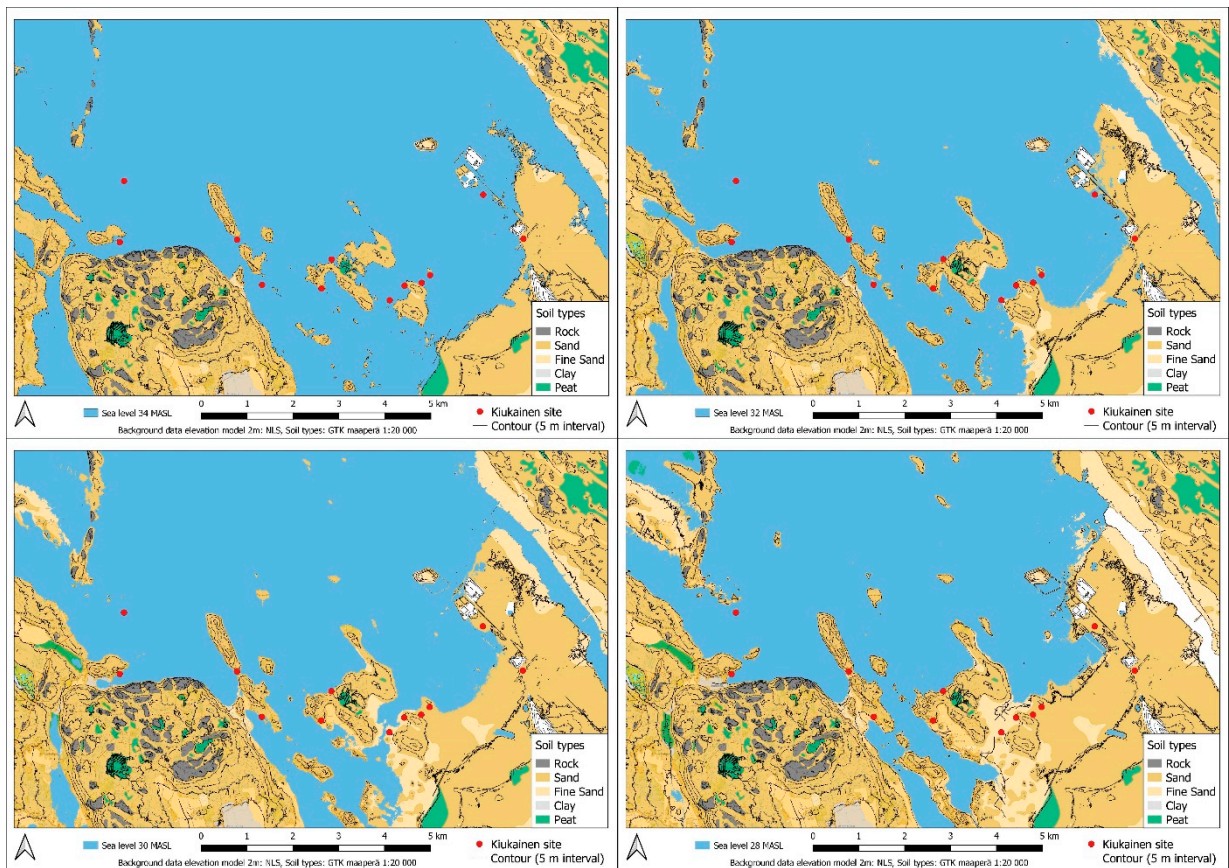

**Figure 8.** The locations of Kiukainen sites in the Harjavalta area with different landscape visualizations. The Sea is visualized at 34 MASL (**upper left**), 32 MASL (**upper right**), 30 MASL (**lower left**), and 28 MASL (**lower right**). Soil types are visualized with different colours.

The maps clearly show how the environment changes quickly as the land rises from the sea (Figure 8). It seems likely that, due to such changes, many residences would have soon been located far from the beach and, thus, probably subject only to short-term use. On the other hand, due to the steeper topography, Uotinmäki, Kaunismäki, and Saamanmäki remained constantly close to the seashore, making them habitable from one century to the next. Kuusisto's site remained underwater throughout the Stone Age and only emerged from the water during the Bronze Age. However, Kiukainen pottery has been found at the site, so either the information about the height of the place is inaccurate or Kiukainen-type pottery was, perhaps, used relatively late in the Bronze Age.

The settlements were located in the areas protected from the wind because the ancient sea bay was wide and open. Based on their location, the immediate proximity to the sea was important, and settlement continued for a long time only in places that have remained close to the shoreline. Inhabitation also continued in such places during the Bronze Age, but Lyytikänharju and Kraakanmäki 1–3, were only used while they remained close to the sea. The surroundings of the residences inhabited for a much longer time at Uotinmäki,

Kaunismäki, and Saamanmäki were suitable for early farming already at the time of the Kiukainen culture, for they were situated on lush slopes.

The settlements in the Kristiinankaupunki area have only been excavated at Langängen in 1950 and in Rävåsen in 1994–1999 [2,35]. The area is known for containing a large number of residences belonging to the Kiukainen culture, but due to the research situation, ceramics have only been found in a few (Figure 9). Most of the settlements seem to have been located by the sea, with the water level having been about 40 m higher than today (Figure 10). The same also applies to places where ceramics have not been found. At that time, they were located on the shores of a sheltered bay formed by the mainland and an island on its western side. The Gulf of Bothnia opened to the western side of the area, and the rivers Kärjenjoki and Lapväärtinjoki ran down to the southwestern end of the area. The waters near the residences would have been sheltered and well suited for fishing and catching waterfowl.

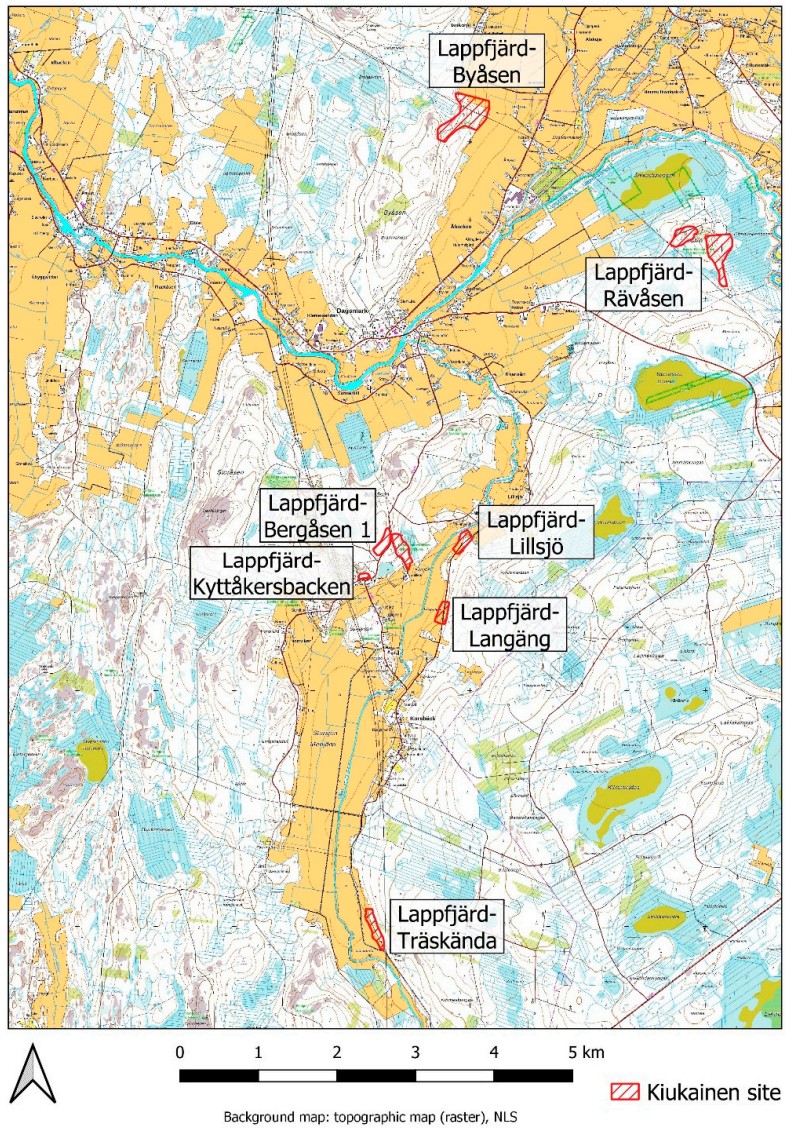

**Figure 9.** The locations of Kiukainen sites in the Kristiinankaupunki area. Grey areas in the background of the map are rocky areas, while the yellow areas are cultivated fields, and the light green or empty white areas in between are forests.

Land uplift in the area occurred quickly at the end of the Stone Age, having been more than a meter per century at the time. The landscape was, therefore, constantly changing, and the sheltered sea area narrowed into two lakes, which were later drained. Their height

would have been about 35 m, but many settlements would have already been far from the shore at this point. In terms of time, the separation of the lakes from the sea dates back to the Bronze Age, approximately 1500 cal. BC. The finds at the settlement called Langäng, which was located on the shore of a smaller lake called Lillsjön, continued to a height of about 35.5 m, and C. F. Meinander, who excavated the site, considers it possible that the settlement continued to be inhabited during the lake phase as well [2]. At other sites, settlement may have continued into the Bronze Age, as several Bronze Age cairns have been found in the area and at the settlement sites. However, the sea connection had already been lost by then, and the settlement's subsistence was probably based on something other than just marine resources.

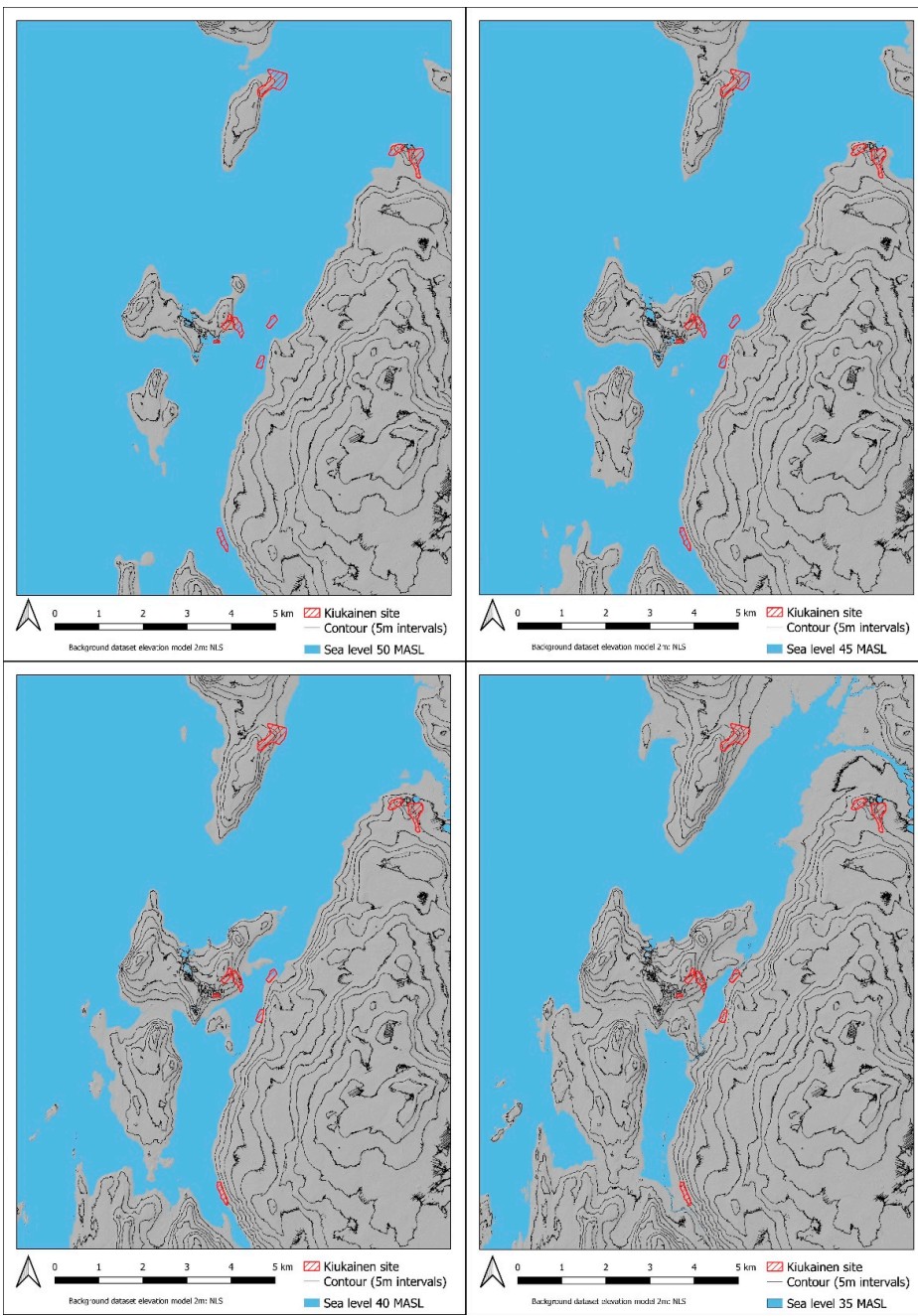

**Figure 10.** The locations of Kiukainen sites in the Kristiinankaupunki area with different landscape visualizations. The Sea is visualized at 50 MASL (**upper left**), 45 MASL (**upper right**), 40 MASL (**lower left**), and 35 MASL (**lower right**).

Rävåsen is a clear exception to the other settlement sites along the shorelines of the sea, having been inhabited for a long time before the Kiukainen culture. However, some Kiukainen ceramics have been found there at a height of approximately 50 m above the sea level today [36]. At the end of the Stone Age, the settlement was located at least four hundred meters from the sea and the mouth of the river Lapväärtinjoki. The area between it and the sea consisted of low reeds and possibly meadows. The site may, therefore, have been used during the time of the Kiukainen culture more for the purposes of tending livestock and engaging in small-scale farming than for taking advantage of marine resources. The surroundings of other sites in the area could also have been suitable for small-scale farming in addition to fishing, as low, seaside meadows and fertile soil would have existed in the vicinity, especially at the very end of the Kiukainen culture.

It must be noted that many more sites in the area around Kristiinankaupunki have been defined as Stone Age settlement sites in the register (Figure 11). More Kiukainen culture sites may exist in the area, but the lack of field studies, and especially excavations, make it difficult to interpret just which of the sites may have been inhabited simultaneously or by the same culture. As seen from the previous map (Figure 8), the seven known Kiukainen sites are located at different heights, and some are multi-period sites. Landscaping and building activities have also damaged some of the sites, so the original site location and zone may have been different than how it appears in the register today.

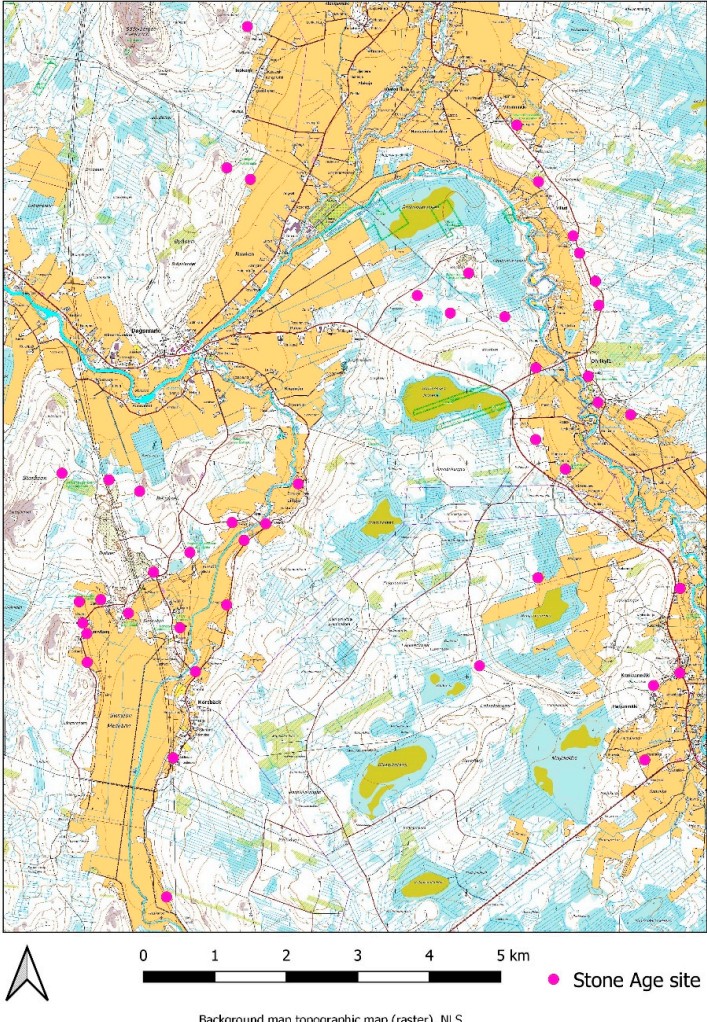

**Figure 11.** The locations of sites that have been defined as Stone Age in the Kristiinankaupunki area. Grey areas in the background of the map are rocky areas, while the yellow areas are cultivated fields, and the light green or empty white areas in between are forests.

As seen from the comparison, the Kiukainen culture sites have been located in variable landscapes and topographies. Easy connection to the sea and landing possibilities have been one key demand for such sites. In the area around the rocky island of Kemiönsaari, the best places have been located on the eastern shores of the island, which would have given the best shelter from the open sea to the west or southwest. In the Harjavalta area, changes in the landscape and location of the seashore occurred rapidly, so some of the sites have been used only for a short period. In the Kristiinankaupunki area, sites are also oriented towards the east or, then, located in a sheltered bay, as the sea opens to the west. With respect to future research and, especially, GIS analysing methods, such great variation in landscapes makes it difficult to use f.x. predictive modelling. The results from the site environment confirm previous knowledge about Kiukainen culture having been a marine culture with certain local adaptations, such as possible small-scale cultivation at some sites.

## 4. Discussion

The adoption of farming and pastoralism in Finnish Neolithic cultures has been a topic of discussion for a long time. According to the latest research, communities in southern Finland engaged in farming and pastoralism even before the Kiukainen culture. Based on pollen studies and the location of the settlements, it has been suggested that small-scale farming may have been important, already, at the time of typical Comb Ceramic culture (4100–3550 cal. BC) [37]. At the latest, these subsistence strategies arrived from elsewhere in the Baltic region together with the Corded Ware culture, from 2900 BC onwards. So far, archaeologists have only found evidence of nomadism practiced in Finland at the time, but traces of dairy fats have been found in pottery [38]. In addition, goat hair has also been found in a grave dated to the time of the Corded Ware culture [39]. However, no evidence of cultivation has been discovered, though it would have been entirely possible based on the location of the settlement sites in fertile environments. The Pyheensilta group has been poorly studied, but based on the location of the settlement sites, marine fishing and hunting were of great importance to such communities. The Pyheensilta group, however, had active connections with groups belonging to the East Swedish Pitted Ware culture in the Åland Islands, where carbonized grains have been found [40].

The subsistence patterns of Kiukainen culture were based, mainly, on marine resources, but evidence of small-scale farming has also been found in recent years. In the excavations done at the Riihivainio settlement site in Turku, archaeologists found evidence of contemporaneous field cultivation in connection with the Kiukainen culture [41]. Most of the excavated settlements have been interpreted as places mainly related to hunting, but archaeologists have discovered grinding stones, especially in the settlements located at the mouth of Kokemäenjoki River, with the stones probably having been used to grind grains [42]. However, all the grinding stones have been found in settlements that continued to be used in the Bronze Age. Farming possibly also included the use of arrow-bladed stone axes, most likely used as hoes [42]. In the distribution area of the Kiukainen culture, signs of cultivation have also been found in the sediments of lakes and moors dating to the end of the Stone Age [29,43,44].

The Kiukainen culture exhibited inhabitation practices in the Åland Islands after the disappearance of the Pitted Ware culture from the same settlement sites. The radiocarbon dates suggests that domesticated animals, such as cattle, sheep, and pig, were kept in the Åland islands during the Late Neolithic by the Pitted Ware culture [45]. However, the cultural and populational continuity between the Pitted Ware culture and the Kiukainen culture is still unclear, but the Kiukainen population may well have also maintained small-scale cultivation and husbandry in the Åland Islands. In addition, the oldest sheep bone found in Finland (2200–1950 cal. BC) comes from one of the northernmost Kiukainen culture sites in Kvarnabba Pedersöre [46]. On the other hand, it has been suggested that Kiukainen culture returned to the hunter-gather-fisher lifestyle [38,47], but based on this evidence, the small-scale husbandry was likely one part of the subsistence on the coastal life.

The settlements belonging to the Kiukainen culture are all concentrated on the shores of the Baltic Sea. Pottery spread inland to only a few places, and they are all in the area of the Kokemäenjoki River watershed. The strong connection of the entire cultural phase to the coasts and archipelagos tells not only of the importance of the sea as a source of food but also about its importance in connecting people between different regions. Without the sea and the archipelago, the Kiukainen culture, with its maritime lifestyle, would never have flourished. The contacts between the settlement sites occurred via water, and such contact must have occurred frequently because the material culture of the various settlements has been quite similar throughout the Kiukainen culture area. Ceramics produced by other contemporaneous cultures have not been found in the sites belonging to the Kiukainen culture area except in the Åland islands, and in this sense, the contact between the inland areas and places along the long coastline seems to have been limited. On the other hand, the material culture shows clear Scandinavian influences, so connections existed across the sea. In the future, it would be important to study those cross-sea contacts in the direction of Scandinavia and the Baltics. The length of the coastline where Kiukainen sites have been found is approximately 720 km, and the length of the core area is approximately 400 km. The distance from the Åland Islands to the nearest coastal sites in Turku or Kemiönsaari is approximately 120 km. In the future, different GIS methods, such as least-cost past analysis, could give interesting results about routes, travel times, and so forth.

However, from the perspective of current research and GIS analysis, the Register of Ancient Monuments has many problems. The level of information and site descriptions vary. In some cases, it may mention pottery type or, for example, dating, but some sites only receive brief descriptions without any important accompanying details. Information about the sites has been collected for decades, and some descriptions or locations can be based on very old surveys or small-scale excavations. The user must evaluate data reliability for each individual site, and thus, forming a reliable overall picture is difficult. A lack of proper classifications or keywords makes the register difficult to use with GIS programs. The points or polygons have age classes, such as dating = "stone age" and type = "settlement site," but they fail to provide any additional search options or keywords; hence, the few existing options do not yield a good result when trying to find more specific information on a site other than just dating or type. Additionally, sites can have similar names, so the only reliable identifier is the individual site number. However, if a user wants to list multiple sites, as in this study, searching each site on a case-by-case basis, using only the site number, is a slow process.

Today, archaeological fieldwork in Finland includes detailed archaeological fieldwork guidelines and instruction [48], updated by the Finnish Heritage Agency. However, the information collected fifty or a hundred years ago is a different story. Conducting GIS analyses with unreliable GIS data is problematic. In the future, better tools to evaluate the data quality will be needed. Adding more tools and search options or keywords could support researchers and authorities, too. Updating the register and collecting new information by doing fieldwork is an ongoing and slow process, and at the same time, storing the data requires new solutions [49].

## 5. Conclusions

The lifestyle of the Kiukainen culture settlements seemingly included a combination of marine resources, seafaring, and small-scale farming, if possible, while being integrated with the local environment and landscape. The use of multiple resources afforded the coastal communities more stable lifestyles in that period of changing climate and environment. However, much more research is still needed, and we would like to open a discussion about how the Kiukainen culture fit into the Stone Age lifestyle and landscape archaeology more generally.

**Author Contributions:** Conceptualization, J.S. and J.R.; methodology, J.S. and J.R.; validation, J.S. and J.R.; formal analysis, J.S.; investigation, J.R.; resources, J.S. and J.R.; writing—original draft preparation, J.S. and J.R.; writing—review and editing, J.S. and J.R.; visualization, J.R.; supervision, J.S. and J.R.; project administration, J.S. and J.R.; funding acquisition, J.S. and J.R. All authors have read and agreed to the published version of the manuscript.

**Funding:** This research was funded by KONE FOUNDATION, grant number 202006680, and NORDENSKIÖLD-SAMFUNDET. Open access funding provided by University of Helsinki.

**Data Availability Statement:** Data available in a publicly accessible repository that does not issue DOIs Publicly available datasets were analyzed in this study. This data can be found at: https://www.museovirasto.fi/fi/palvelut-ja-ohjeet/tietojarjestelmat/kulttuuriympariston-tietojarjestelmat/kultt uuriympaeristoen-paikkatietoaineistot (accessed on 20 August 2022) and https://aland.maps.a rcgis.com/apps/webappviewer/index.html?id=9d7cc07ab4004f0ca620038c4fd416ca (accessed on 20 August 2022).

**Conflicts of Interest:** The authors declare no conflict of interest.

## Appendix A

**Table A1.** The list of archaeological sites where Kiukainen pottery has been found and discussed in this study.

| Municipality | Site Number | Site Name |
|---|---|---|
| Espoo | 49010002 | Finns |
| Espoo | 49010040 | Mynt |
| Espoo | 49010004 | Backisåker 1 |
| Espoo | 49010001 | Grankulla |
| Espoo | 49010021 | Lillgus storåker |
| Eurajoki | 51010029 | Irjanteen hautausmaa |
| Eurajoki | 51010009 | Etukämppä |
| Hamina | 917010013 | Hietojanvuori |
| Harjavalta | 79010009 | Kuusisto E |
| Harjavalta | 79010001 | Kaunismäki A ja B |
| Harjavalta | 79010006 | Saamanmäki |
| Harjavalta | 79010025 | Lyytikänharju |
| Harjavalta | 79010023 | Sievarintie E |
| Harjavalta | 1000038606 | Kraakanmäki 3 |
| Harjavalta | 1000038607 | Kortteenrapakko |
| Harjavalta | 1000006682 | Hakala |
| Harjavalta | 1000022768 | Kraakanmäki 2 |
| Harjavalta | 1000022767 | Kraakanmäki 1 |
| Humppila | 103010001 | Järvensuo 1 |
| Inkoo | 1000000046 | Malmskyan |
| Inkoo | 149010060 | Vahrs |
| Inkoo | 149010056 | Malmgård |
| Inkoo | 149010068 | Staffans |
| Inkoo | 149010069 | Nysvenskas |
| Inkoo | 1000006090 | Kasabergen |
| Kaarina | 202010026 | Ravattula Ristimäki |

**Table A1.** *Cont.*

| Municipality | Site Number | Site Name |
|:---:|:---:|:---:|
| Kaarina | 202010020 | Muikunvuori |
| Kangasala | 211010006 | Sepänjärvi 1 |
| Kemiönsaari | 243010042 | Nedergård |
| Kemiönsaari | 243010045 | Eländet |
| Kemiönsaari | 1000019364 | Ölmosviken |
| Kemiönsaari | 40010014 | Knipnäsbacken |
| Kemiönsaari | 40010036 | Hammarsboda 4 |
| Kemiönsaari | 40010015 | Jordbro |
| Kemiönsaari | 1000031089 | Näset (Skinnarviksvägen) |
| Kirkkonummi | 257010027 | Pappila |
| Kirkkonummi | 257010053 | Framhoparn |
| Kirkkonummi | 257010081 | Kolsarby |
| Kotka | 285010017 | Niskasuo |
| Kristiinankaupunki | 409010030 | Lappfjärd-Bergåsen 1 |
| Kristiinankaupunki | 409010049 | Lappfjärd-Träskända |
| Kristiinankaupunki | 409010047 | Lappfjärd-Lillsjö |
| Kristiinankaupunki | 409010028 | Lappfjärd-Langäng |
| Kristiinankaupunki | 409010040 | Lappfjärd-Kyttåkersbacken |
| Kristiinankaupunki | 409010041 | Lappfjärd-Byåsen |
| Kristiinankaupunki | 409010044 | Rävåsen |
| Laitila | 1000000142 | Ahtkorvenmäki |
| Laitila | 1000000092 | Hangassuo |
| Laitila | 1000004424 | Miilunpohjansuo |
| Lohja | 444010047 | Kittiskoski E |
| Loppi | 433010014 | Kuitikas |
| Loviisa | 701010023 | Koirankallio |
| Loviisa | 585010015 | Strömbo |
| Mynämäki | 503010040 | Pyheensilta, Laajoen luoteispuoli |
| Nakkila | 1000001335 | Uotinmäki ja Uotinmäki W |
| Nakkila | 531010005 | Kuusisto |
| Nousiainen | 538010037 | Kylävuori |
| Närpiö | 605010001 | Pörtom-Raineåsen |
| Närpiö | 605010020 | Pörtom-Langbacken |
| Paimio | 577010030 | Kehioja |
| Paimio | 577010038 | Halkilahti |
| Pedersöre | 990010035 | Esse-Jättegobacken/Smedasforsen A + B |
| Pedersöre | 990010004 | Esse-Kvarnnabba |
| Pori | 609010084 | Kirkkokangas IV |
| Porvoo | 613010040 | Böle |

**Table A1.** *Cont.*

| Municipality | Site Number | Site Name |
|---|---|---|
| Pyhtää | 624010017 | Brunamossen 2 |
| Pyhtää | 624010029 | Trollberget |
| Pyhtää | 624010037 | Eetinniitty 1 |
| Pyhtää | 624010036 | Kaarlinsaari 1 |
| Pyhtää | 624010038 | Eetinniitty 2 |
| Pyhtää | 1000007139 | Nygård 2 |
| Pyhtää | 1000007141 | Längkärrsskogen 2 |
| Pyhtää | 1000007159 | Nygård 1 |
| Pyhtää | 1000007149 | Eetinniitty 4 |
| Pyhtää | 1000016854 | Eetinniitty 5 |
| Raasepori | 220010082 | Grågälan-Träskhusåkern |
| Raasepori | 220010036 | Dragongatan |
| Raasepori | 1000039580 | Sannäsmalmen |
| Raasepori | 1000032776 | Gloviken |
| Salo | 734010002 | Alhonpelto |
| Sastamala | 912010046 | Liekolankatu |
| Sastamala | 912010022 | Haapakallio |
| Sastamala | 912010017 | Hiukkasaari |
| Seinäjoki | 975010014 | Viinapränninlaakso |
| Turku | 853010022 | Kotirinne |
| Turku | 202010037 | Pähkinämäki 2 |
| Turku | 853010008 | Riihivainio |
| Turku | 853010048 | Niuskala |
| Turku | 853010019 | Maaria Kärsämäki |
| Turku | 853010029 | Kanttorinmäki |
| Ulvila | 293010007 | Eskola |
| Ulvila | 293010006 | Hämäläinen I |
| Virolahti | 935010004 | Kattelus 1 |
| Vöyri | 559010022 | Torplindorna S |
| Vöyri | 559010018 | Fårmossen 1–2 |
| Vöyri | 559010025 | Torplindorna N |
| Åland-Saltvik | Sa 20.8 | Myrsbacka I |
| Åland-Saltvik | Sa 21.1 | Krokars |
| Åland-Saltvik | Sa 20.8 | Svinvallen |

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
