# Peer review of "Kiukainen Culture Site Locations—Reflections from the Coastal Lifestyle at the End of the Stone Age"

_land, doi:10.3390/land11091606_

Round 1

Reviewer 1 Report

Please see attached review.

Author Response

Thank you for your supportive comments and good editing suggestions. Instead of reversing figures 1 and 2, we have added one new figure to the introduction part where Southern Finland is highlighted, and rest of the Europe is present. We hope that this is ok for you and this new map helps readers to understand the geographical context from the very beginning.  

It is a definitely good idea to give numbers to the sites listed in the table and we have considered it. However, to get all those 99 numbers to the distribution map would make it very messy or another option would be many new maps on a different scale. Since we are mostly trying to visualize the whole big area where culture was active, we think that single site names or numbers don’t need to be added. All exact site locations and coordinates can be found fx. from the Finnish Heritage Agency webpage (Muinaisjäänösrekisteri) by using the municipality and site name that is presented in the table. In the future, we will keep this in mind and try to find a solution for how to visualize a large number of sites so that numbers can be used.  

Thank you also for all the English corrections. Our paper was proofread by a professional expert (the University of Helsinki supports that) but it was done in a hurry and he clearly missed some of our mistakes.  

It is an interesting idea to list Stone-Age periods on a table (or maybe a nice illustration) and we will definitely use this idea in the future. Unfortunately, there are still some time periods that need more research and especially datings, especially at the end of the Stone Age, before the full timeline could be reconstructed. There are of course some well-known phases in the Finnish Stone Age, but still, a lot of research needs to be done. We hope that in the near future we will know much more about the Kiukainen culture time period and also the cultural phases before it. 

Reviewer 2 Report

Please add the results of C14 datings even just in a table if possible 

Author Response

Thank you for your review. Dating of the Kiukainen culture is still a little bit unclear and needs more research. We have only a few C14 datings from the Kiukainen culture and almost all of them are unpublished. I will come back to this dating in my next article which deals with the Kiukainen pottery and new C14 datings.

Sincerelly

Janne Soisalo

Reviewer 3 Report

Dear authors,

I found the manuscript interesting and I will be very pleased to see it published in its final form in the journal Land.

I have small remarks concerning the way work is presented that I would like to see considered in a final version of the manuscript.

#1 Location figure (Fig. 2)

Figure 2 locates the Kiukainen sites in South Finland, though it can be difficult to understand for international readers; I suggest to add to this figure a context map with N Europe. The figure title should also include a reference to the location in South Finland. 

#2 Figures captions (Figs. 2 to 10)

The titles (captions) of figures should be revised, as these must be the most complete as possible. They should be legible without the need to see the main text. For instance, some of these figures captions do not refer the study area or the study context. In some of them, these data are imbedded in the figure, as a title, what should be removed from all of them. For instance, in the figures with MASL references, these data must be included in the figure title (caption) and removed from the images.

#3 Sites pictures  

It would be interesting to have another figures with photographies of the sites (at least some) where the pottery was found, to have a landscape contextualization.  

#4 Sea level interpretation

It would be interesting to have more information on the sea level variation in during the last 5 ka and its relation with the region’s specificities. That could be done in the Discussion chapter. For instance, the authors refer that the land uplifted more than a meter per century at the end of the Stone Age (lines 333-335), but it is important to contextualize this phenomenon (in counter-cycle with sea level rise in other regions of Europe) in the scope of isostatic compensation due to deglaciation in northern Europe after the Last Glacial Maximum.

Regards

Author Response

Thank you for your supportive comments and good editing suggestions. We have added one new figure to the introduction part where Southern Finland is highlighted, and rest of the Europe is present. We hope that this new map helps readers to understand the geographical context from the very beginning.

It is an excellent idea to remove the titles from the images and revise the figure captions! All titles are now removed from the images and figure captions are supplemented. In Finland, it is maybe just an old custom to add map titles and other details inside the figure.

Unfortunately, it is quite difficult to take a landscape photograph from the sites mentioned in this article because most of the sites are located in forest areas or scrubs where landscape can’t be seen. Also, the build environment and cities don’t give a good overview of the landscape. However, we agree that it is important to have a landscape contextualization and for future articles, we should consider maybe drone photographing or other solutions.

The land uplift varies in Finland, being the fastest in the west. Its speed has remained almost the same for thousands of years, but it slows down towards modern times. The Kiukainen culture covers a large area and the differences between the different settlement sites are explained on the maps. However, land uplift has not been studied in great detail and we feel that it is more of a field of research for geologists. It is difficult to get accurate calculations and maps have been used to visualize the issue.